# PROPENSITY GUIDED TRANSFORMER FOR CAUSAL EFFECT INFERENCE

## ABSTRACT

We introduce the Propensity Similarity guided Bidirectional Transformer (PSBT), a novel framework designed to estimate causal effects in observational data while addressing confounding bias. PSBT employs a pre-training and fine-tuning approach to learn causal representations, guided by propensity scores. In the pre-training phase, the model predicts masked covariates (self-supervised learning) and propensity similarity between unit pairs (weakly supervised learning), enabling the representation space to disentangle confounding factors. The fine-tuning stage leverages these representations for causal outcome prediction, refining them for counterfactual reasoning. Experiments on multiple benchmark datasets demonstrate that PSBT significantly outperforms traditional and state-of-the-art causal inference methods in estimating the Conditional Average Treatment Effect (CATE) and other metrics. By emphasizing propensity-guided learning over conventional balancing techniques, PSBT achieves robust and interpretable representations, advancing deep learning model capabilities in causal effect inference tasks.

## 1 INTRODUCTION

With the rapid developments of artificial intelligence in wide areas, it is highly needed that deep learning models should have the capability of reasoning, e.g., logic reasoning or mathematical reasoning. Answering questions like "*What is cause?*" and "*What is effect?*" from the observational data is regarded as the initial step to build general artificial intelligence Pearl (2019). Properly answering questions like "*What would the patient's health condition be had they received medication A?*" is the central concern of causal effect inference. These studies greatly benefit research in healthcare Casucci et al. (2018; 2019), educational studies Zhao & Heffernan (2017), economics policy making Smith & Todd (2005); Lalonde (1984) and sociology Morgan & Harding (2006). Although most of the research for causal inference is based upon Randomized Controlled Trials (RCTs) Bertsimas et al. (2019), there are a lot of attempts focusing on observational data, also known as observational studies Rosenbaum (2002). The most important challenge for causal inference in observational studies is to tackle confounding bias in the data collection process, where confounders affect both the effects of intervention variables on the outcome variables, and the intervention variables themselves. Techniques to mitigate confounding bias in causal inference range from covariate matching based optimization methods Stuart (2010) to regression correction based statistical methods Chipman et al. (2008). Recent advances in domain adaptation suggest that a well constructed representation learning model could improve the performance of counterfactual reasoning model significantly on multiple benchmarks Johansson et al. (2016); Du et al. (2019). However, matching-based methods require that all the confounders should be able to be measured, so that the information from treatment variable to response variable could be blocked according to the back-door principles Pearl (2010). In scenarios where only noisy and dependent proxy variables are available, latent variable methods are needed to recover the true confounders for causal inference Louizos et al. (2017).

We explore a new framework to estimate causal effect under confounding bias. In this framework, we use a bidirectional transformer model to learn the feature representations for the covariate features. The representation learning framework is guided by two tasks. On the one hand, we randomly mask the covariate features, and formulate a self-supervised task to predict the masked covariate features. On the other hand, we combine two units to formulate a subsequent propensity similarity

prediction task, where we use the propensity score model to guide the unit pairs as their supervision for propensity similarity prediction. Hence, we are trying to distill the propensity knowledge into the feature representation learning model. Rather than learning a balanced representation between treatment and control groups, this method aims to learn the propensity model in the pre-training stage.

In addition, we formulate a fine-tuning task based on the pre-trained propensity similarity prediction modeling stage. In the fine-tuning stage, an additional layer of a neural network is formulated to enable the causal outcome prediction task, where we fine-tune the learned representations as shared input to the causal prediction layer for the final causal effect inference task.

In order to teach the model how to learn the differences between covariate pairs, rather than performing balancing, we propose to employ the propensity score as knowledge guidance for the attention mechanism to jointly aggregate the covariate representations. Rather than learning a balanced representation and performing counterfactual inference, we aim to teach the model to learn the propensity. Thus, the representation space is able to disentangle the confounding factors, which would help the counterfactual predictors learn the correct outcome patterns by satisfying the strong ignorability condition.

## 1.1 MAIN CONTRIBUTIONS

- We explore propensity difference prediction as a task of transformer model for learning causal representation.
- We leverage self-supervised learning as masked tokenization to restrict the latent representation while learning the propensity model.
- We perform fine-tuning with causal effect inference task from the pre-trained representations, which allows for causal modeling of treatment effect.
- We build a new framework called PSBT by employing the **P**ropensity **S**imilarity guided **B**idirectional **T**ransformer model, using a pre-training fine-tuning regime. Various experiments on multiple datasets have shown that our proposed PSBT could achieve great performance in causal effect inference tasks. Source code for reproducing our experiments are released for reviewing purposes `https://anonymous.4open.science/r/PSBT-635C/`.

## 2 PROBLEM SETUP

Given a dataset $\mathbb{D} = \{X, T, Y\}$, where $X \in R^{n \times k}$, and $Y \in R^n$, and $T = \{0, 1\}^n$. We have covariate $X_1, \cdots, X_k$, where for each unit $x$, an interventional variable $t$ is assigned and the factual outcome of that intervention is $y_f$. According to the Rubin-Neyman causal model Rubin (2005), for $t \in \{0, 1\}$, we have a joint distribution $P_x = (x, t, y_0, y_1)$. Here $y_0, y_1$ represent the factual and counterfactual outcome, respectively, regarding to $t$ as $y_t, y_{1-t}$. Our target is to learn a model to infer the potential outcome according to the interventional variables.

Unlike the variational auto-encoder-based deep latent model family, we explore the transformer-based model family by self-supervised learning. We show that by properly designing the tasks with a transformer structure, the learned latent representation is able to gain some causal representation features useful for downstream causal inference tasks.

A directed graph model could be used to represent the relation between latent variables and the observable variables in Bayesian formulas, which enables the discovery of true causal factors by posterior approximation Schölkopf et al. (2013). On the other hand, propensity score matching satisfies the strong ignorability condition: it hence indicates the causal direction which aligns with the directed graphic model Rosenbaum (1996; 2002). We show that by pre-training with contrastive framework under propensity guidance, and fine-tuning with causal effect prediction tasks, the transformer model could be capable of conducting causal effect inference tasks.

On the one hand, self-supervised learning maps the input covariates to the representation space with implicit regularization of mutual information, so that the distribution of latents are formulated. On the other hand, either an additive noise model or a variational auto-encoder requires to model the latents in a Gaussian prior, which aligns with the propensity score interpreted as probit function.

Hence, the propensity guidance for a representation learned from the transformer framework could be equivalent to the directed graphic model.

In order to make causal inference identifiable via observational data, we make the following assumptions:

**Assumption 1** *(Strong Ignorability) Conditioning on $x$, the potential outcomes $y_0, y_1$ are independent of $t$, which can be stated as:* $(y_0, y_1) \perp\!\!\!\perp t|x$.

**Assumption 2** *(No Interference) The treatment outcome of each individual is not affected by the treatment assignment of other units, which can be formulated as:* $Y^u(t^1, \cdots, t^n) = Y^u(t^u)$.

**Assumption 3** *(Consistency) The potential outcome $y_t$ of each individual is equal to the observed outcome $y$, if the actual treatment received is $T = t$, which can be represented as:* $y = y_t$, *if* $T = t, \forall t$.

**Assumption 4** *(Positivity) For all sets of covariates and for all treatments, the probability of treatment assignment will always be strictly larger than 0 and strictly smaller than 1, which can be expressed as:* $0 < P(t|x) < 1$, $\forall t$ *and* $\forall x$.

Here, Assumption 1 indicates that all the confounders can be measured so that the confounders can all be controlled for the adjustment to remove the bias. This is a restrictive but much used assumption in a large subset of causal inference literature Rosenbaum & Rubin (1983). Assumption 4 allows us to estimate the treatment effects for any $x$ in the covariate space. With these assumptions, we can formalize the definition of the CATE as follows:

**Definition 1** *The Conditional Average Treatment Effect (CATE) for unit $u$ is:* $CATE(u) := \mathbb{E}[y_1|x^u] - \mathbb{E}[y_0|x^u]$.

This definition restricts the conditional probability as the formal definition of individual level causal effects. Now we can define the *Average Treatment Effect* (ATE) and the *Average Treatment effect on the Treated* (ATT) as:

**Definition 2** *ATE* := $\mathbb{E}[CATE(u)]$, *ATT* := $\mathbb{E}[CATE(u)|t = 1]$.

Here, since the counterfactual outcome cannot be known, we do not know the joint distribution $P(x, t, y_0, y_1)$. We can only estimate a function over the covariate space $\mathbf{X}$ which is defined as $f : \mathbf{X} \times \{0, 1\} \to \mathbf{Y}$. The estimate of CATE($u$) can now be defined as:

**Definition 3** *Given a dataset $\{X, T, Y\}$ and a function $f$, for each unit $u$, the estimate of CATE(u) is:* $\widehat{CATE}(u) = f(x^u, 1) - f(x^u, 0)$.

Our main aim is to learn a proper function to approximate this quantity.

## 3 METHODOLOGY: PSBT

Our framework **PSBT**: **P**ropensity **S**imilarity guided **B**idirectional **T**ransformer consists of two steps when training: pre-training and fine-tuning. During the pre-training step, the model is trained to distinguish the propensity similarity on the covariate feature pairs, without accessing the outcome variables. In this step, a propensity model parameterized by a neural network is first learned on the covariate features with supervision of the treatment assignment. Then, this model is used to supervise the covariate feature pairs to enable the propensity guidance. During the fine-tuning step, the model is initialized with the whole parameters in the pre-training step. Here the treatment outcome variable is used as the regression target, with an additional layer in the pre-training model as the component for the prediction task. In the whole process, our model uses a unified architecture design and there is little difference between the pre-training and fine-tuning downstream tasks.

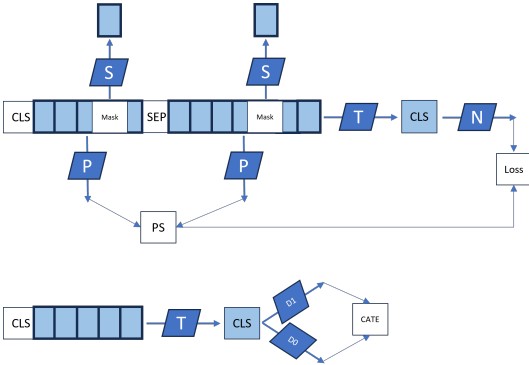

Figure 1: The model framework for the proposed method. In the upper figure, S represents the self-supervised learning component that predicts masked tokens. P represents the propensity component that predicts the propensitys scores. T represent the pretrained target that predicts the propensity similarities. N represnts the next similarity prediction component that formulates the target loss. In the lower figure, T represnts the target prediction that predicts the cls value, D1 and D0 are two causal prediction head that predicts tha causal quantities.

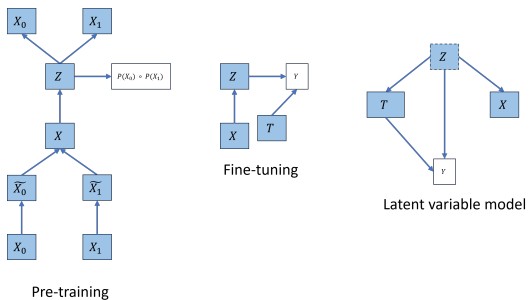

Figure 2: The graphic model for the proposed method.

**Model Architecture** The basic architecture of PSBT is a multi-layer bidirectional transformer encoder that encodes the covariate features into latent representations. This architecture is based on the implementation described in Vaswani et al. (2017); our implementation is based on the Pytorch library. The official implementation of this architecture has been released in the tensor2tensor library[1]. We denote the number of layers, the transformer blocks as L, the hidden size as H, and the number of self-attention heads as A. The report is based on a model PSBT(L=8, H=192, A=8). For the fine-tuning step, we keep most of the architecture the same and add two additional layers to model the treatment outcomes from different interventional arms separately. Figure 1 displays the details of our architecture.

**Input / Output Representations.** As shown in Figure 2, our method firstly projects the input features to token spaces, where a token is represented by two or three features. A single-layer network is used as the project function to transform the input features to the sequential tokens. For the pre-training stage, a CLS token is initialized to concatenate with the sequential tokens, the final hidden state of representation corresponding to this token is used for the prediction tasks. Unit pairs are packed together into a single sequence, where sequences of unit features are separated by a single token Sep, and a learned embedding is added to every single token to indicate whether or not the feature token belongs to the first or second unit. Positional encoding tokens are also included to add to each token as the formulation of the final input representations to the transformer blocks. For a given token, the final state of that token in the input space is summed by the corresponding token, the segment and position embeddings.

---

[1]https://github.com/tensorflow/tensor2tensor

## 3.1 Pre-training Causal Representation

Our architecture is used to encode the bidirectional sequential feature to pre-train PSBT. We train PSBT with one weakly-supervised task and one self-supervised task, as illustrated in Figure 1.

**Predict the propensity similarity.** The main goal of this step is to let the model understand the relation between the unit pairs, regarding the propensity similarity. In order to guide the transformer model with the knowledge of propensity scores, this framework sets up a classification token by a learning target to predict the propensity similarity. Back-propagation of errors from neural networks output to the real-world annotation promotes the success of modern deep learning methods. The model framework in this paper makes use of back-propagation from propensity similarity to guide the transformer learning the causal representation for downstream inference tasks. Propensity guidance allows for the representation learning to group the units within the same stratification into a local manifold, so that treatment outcomes from both domains can be learned by the neural networks function, which equals to the adjustment of latent variables.

We obtain the propensity score for each unit by using a pre-trained neural network, specifically: a multi-layer fully connected neural network with BatchNormalization and ReLU layers. This network is trained on the tasks with input as the feature vectors and output as the treatment assignment: given a unit with covariate features, the network outputs the propensity of treatment assignment. This pre-trained network generates propensity similarity for each unit pair.

Another issue to be considered in this method is the choice of metric for measuring the propensity similarity. The output of nuisance functional model maps the covariates to the logit space, representing the similarity between propensities. In order to formulate a stable target space for the model to learn the propensity similarity, this method uses soft binary cross entropy to model the prediction error, by projecting the logit space into the probability space scaling from zero to one. An alternative method is to model the output space with softplus function so that the value can be restricted to $\mathbb{R}^+$.

**Self-supervised covariates feature regression.** In order to train a deep bidirectional representation, we mask some percentage of the input tokens at random, and then predict those masked tokens. Here, each token consists of feature compositions, for which real-valued features are projected into real value spaces. This step is the same as the 'masked language model' but the sequential part is not sentence sequences any more, but covariate features. The MASK tokens are transformed into latent representations in the final hidden state of the transformer blocks and as the input of a final regression layer. The ground truth feature values are used as the supervised signal to test the self-supervised feature regression task. In order to mitigate the covariate shifts between pre-training and fine-tuning, we do not always use a MASK token to replace the tokens. There are 10% positions of token are marked by the MASK token, and among them 80% are kept with the masked condition, 10% are randomly replaced by other tokens, and 10% of the them remain the same.

In conclusion, the framework of our proposed method as illustrated in Figure 1 encompasses a modeling process with two steps. First, pre-training is conducted by introducing the propensity model for sequence level supervision and self-supervised learning is conducted to regulate the learned representations. Subsequently, a task level supervision is conducted to enable the network to do counterfactual reasoning.

## 3.2 Fine-tuning Causal Model

In the fine-tuning stage we keep all the parameters from the pre-training stage instead of two additional layers, where each layer represents the task of predicting the causal outcomes of the model, as shown in the bottom figure in Figure 1. This step is straightforward due the self-attention mechanism of transformer architectures. In the fine-tuning stage, only one unit is used as inputs to the network and the final hidden state of the CLS token is used as the input to the causal layer. There are two causal layers, each corresponding to one potential outcome $Y_0, Y_1$. The fine-tuning stage enables the model to further adjust the representation learning pre-trained from the last stage to adjust according to each potential outcomes.

Table 1: Statistics on the employed datasets. In the MIMIC-III dataset, the numbers of control and treatment units are simulated. The details of both procedures are provided in the main text, specifically in the corresponding paragraphs of Section 4.1, where each dataset is introduced. Additionally, note that the control unit pool in the Jobs dataset consists of two components (cf. Section 4.1, Jobs paragraph).

| Dataset | Observations | Control/treatment | Covariates | Reference |
|---------|-------------|-------------------|-----------|-----------|
| IHDP | 747 | 608/139 | 25 | Hill Hill (2011) |
| Jobs | 3 122 | 2 915/297 | 7 | Lalonde Lalonde (1984) |
| MIMIC-III | 7 413 | -/- | 25 | Johnson et al. Johnson et al. (2016) |
| Twins | 25 656 | 12 828/12 828 | 43 | Louizos et al. Louizos et al. (2017) |

## 4 EXPERIMENTS

Our experiments aim to answer the following questions:

- How effective is the Transformer-based model in learning representations for causal inference tasks?
- How much benefit could the propensity supervision bring for the causal inference tasks?

### 4.1 DATASETS

To properly assess our method, we run experiments on several datasets that are designed to evaluate causal effect inference tasks. Because of the unobservable counterfactual outcomes, semi-simulated or simulated datasets are used to create ground truth data Hill (2011). Table 1 lists summary statistics for the datasets.

**IHDP**    Hill (2011). The Infant Health and Development Program (IHDP) examines the effect of specialist home visits on infants' future cognitive test scores. This semi-simulated dataset is derived from covariates collected during a real-world randomized experiment. Treatment selection bias is introduced by excluding a subset of the treatment group. Treatment outcomes are simulated using Setting 'A' as described in Dorie (2016). The dataset includes 747 units: 608 in the control group and 139 in the treatment group, with each unit characterized by 25 covariates.

**Jobs**    Lalonde (1984); Smith & Todd (2005). The Jobs dataset evaluates the impact of job training on employment outcomes. It combines a randomized component from the National Supported Work program with a non-randomized component from observational studies. The randomized dataset includes 722 units (425 control and 297 treated) with seven covariates. The non-randomized dataset (PSID comparison group) consists of 2 490 control units.

**MIMIC-III**    Johnson et al. (2016). This benchmark dataset is derived from MIMIC-III, a database of de-identified patient profiles and health outcomes for critical care unit patients. The dataset includes demographic details and observed laboratory measurements (chemistry and hematology). After filtering for missing values, the dataset comprises 7 413 samples, each with 25 covariates. The binary treatment examines the effect of prescription amount on ICU length of stay: $t = 0$ represents a small prescription amount, and $t = 1$ represents a large prescription amount. Treatment outcomes are simulated as $y|x, t \sim (w^T + \beta t + n)$, where $n \sim N(0, 1)$, $w \sim N(0, 0.5 \cdot (\Sigma + \Sigma^T))$, and $\Sigma \sim U((-1, 1)^{25 \times 25})$. Treatment assignment follows $t|x \sim Bern(\sigma(s^T x + m))$, where $m \sim N(0, 0.1)$ and $s \sim N(0, 0.1 \cdot I)$.

**Twins**    Louizos et al. (2017). The Twins dataset is constructed from the "Linked Birth/Infant Death Cohort Data" by NBER. Using a matching algorithm, it selects twin births in the USA from 1989 to 1991. The dataset contains 43 covariates, including parental demographics (education, age, race), health factors (prenatal care timing, number of prenatal visits), and other conditions. Only same-gender twin pairs weighing less than 2 000g are included. The treatment variable assigns $t = 0$ for the lighter twin and $t = 1$ for the heavier twin, with the first-year mortality rate as the outcome.

Table 2: In-sample and out-of-sample results with mean and standard errors on the IHDP dataset (lower = better).

| Methods | In-sample | | Out-sample | |
|---|---|---|---|---|
| | $\sqrt{\epsilon_{\text{PEHE}}}$ | $\epsilon_{\text{ATE}}$ | $\sqrt{\epsilon_{\text{PEHE}}}$ | $\epsilon_{\text{ATE}}$ |
| OLS/$LR_1$ | 5.8 $\pm$ .3 | .73 $\pm$ .04 | 5.8 $\pm$ .3 | .94 $\pm$ .06 |
| OLS/$LR_2$ | 2.4 $\pm$ .1 | .14 $\pm$ .01 | 2.5 $\pm$ .1 | .31 $\pm$ .02 |
| S.Learner | 1.7 $\pm$ .6 | .18 $\pm$ .04 | 3.0 $\pm$ .5 | .36 $\pm$ .06 |
| T.Learner | 1.5 $\pm$ .1 | .17 $\pm$ .03 | 2.7 $\pm$ .6 | .33 $\pm$ .04 |
| BLR | 5.8 $\pm$ .3 | .72 $\pm$ .04 | 5.8 $\pm$ .3 | .93 $\pm$ .05 |
| BART | 2.1 $\pm$ .1 | .23 $\pm$ .01 | 2.3 $\pm$ .1 | .34 $\pm$ .02 |
| k-NN | 2.1 $\pm$ .1 | .14 $\pm$ .01 | 4.1 $\pm$ .2 | .79 $\pm$ .05 |
| RF | 4.2 $\pm$ .2 | .73 $\pm$ .05 | 6.6 $\pm$ .3 | .96 $\pm$ .06 |
| CF | 3.8 $\pm$ .2 | .18 $\pm$ .01 | 3.8 $\pm$ .2 | .40 $\pm$ .03 |
| BNN | 2.2 $\pm$ .1 | .37 $\pm$ .03 | 2.1 $\pm$ .1 | .42 $\pm$ .03 |
| TARNet | .88 $\pm$ .0 | .26 $\pm$ .01 | .95 $\pm$ .0 | .28 $\pm$ .01 |
| CFR-Wass | .71 $\pm$ .0 | .25 $\pm$ .01 | .76 $\pm$ .0 | .27 $\pm$ .01 |
| CEVAE | 2.7 $\pm$ .1 | .34 $\pm$ .01 | 2.6 $\pm$ .1 | .46 $\pm$ .02 |
| SITE | .69 $\pm$ .0 | .22 $\pm$ .01 | .75 $\pm$ .0 | .24 $\pm$ .01 |
| ABCEI | .71 $\pm$ .0 | .09 $\pm$ .01 | .73 $\pm$ .0 | .09 $\pm$ .01 |
| PSBT | **.51 $\pm$ .0** | **.03 $\pm$ .01** | **.53 $\pm$ .0** | **.03 $\pm$ .01** |

Mortality is 19.02% for lighter twins and 16.54% for heavier twins. Observational outcomes for both treatments are available. Selection bias is simulated by selectively observing one twin based on covariates, modeled as $t|x \sim \text{Bern}(\sigma(w^T x + n))$, where $w^T \sim N(0, 0.1 \cdot I)$ and $n \sim N(1, 0.1)$.

## 4.2 BASELINE METHODS

We consider three groups of baselines:

1. Statistical estimators: least square regression using treatment as a feature (OLS/$LR_1$); separate least square regressions for each treatment (OLS/$LR_2$); a single network with treatment as covariates (S.learner Künzel et al. (2019)); separate neural regressors for each treatment group (T.learner Künzel et al. (2019)); random forest (RF Breiman (2001)).

2. Matching-based estimators: balancing linear regression (BLR); k-nearest neighbor (k-NN Crump et al. (2008)); causal forest (CF Wager & Athey (2018)); Bayesian additive regression trees (BART Sparapani et al. (2016)).

3. Learning-based estimators: balancing neural network (BNN Johansson et al. (2016)); treatment-agnostic representation networks (TARNet) and counterfactual regression with Wasserstein distance (CFR-Wass Shalit et al. (2017)); causal effect variational autoencoders (CEVAE Louizos et al. (2017)); local similarity preserved individual treatment effect (SITE Yao et al. (2018)) and adversarial balancing-based representation learning for causal effect inference (ABCEI Du et al. (2019)).

We demonstrate a quantitative comparison between our proposed method and the baseline methods. All baseline methods are parameterized according to the recommended settings in the original papers.

## 4.3 EVALUATION METRICS

We use a semi-simulated method to include the benchmark datasets like IHDP and MIMIC-III, so that we can know the ground truth for the CATE estimation. Hence, we can use *Precision in Estimation of Heterogeneous Effect* (PEHE) Hill (2011) as the evaluation metric of CATE estimation:

$$\epsilon_{\text{PEHE}} = \frac{1}{n} \sum_{u=1}^{n} ((\mathbb{E}[y_1|x^u] - \mathbb{E}[y_0|x^u]) - (f(x^u, 1) - f(x^u, 0)))^2.$$

Subsequently, the precision of ATE estimation can be evaluated based on the estimated CATE. On the Jobs dataset, because we combine non-randomized components and randomized components,

Table 3: In-sample and out-of-sample results with mean and standard errors on the Jobs dataset (lower = better).

| Methods | In-sample | | Out-sample | |
|---|---|---|---|---|
| | $R_{\text{pol}}$ | $\epsilon_{\text{ATT}}$ | $R_{\text{pol}}$ | $\epsilon_{\text{ATT}}$ |
| OLS/$LR_1$ | $.22 \pm .0$ | $\mathbf{.01 \pm .00}$ | $.23 \pm .0$ | $.08 \pm .04$ |
| OLS/$LR_2$ | $.21 \pm .0$ | $.01 \pm .01$ | $.24 \pm .0$ | $.08 \pm .03$ |
| S.Learner | $.21 \pm .0$ | $.02 \pm .01$ | $.24 \pm .0$ | $.08 \pm .03$ |
| T.Learner | $.20 \pm .0$ | $.02 \pm .01$ | $.22 \pm .0$ | $.08 \pm .03$ |
| BLR | $.22 \pm .0$ | $.01 \pm .01$ | $.25 \pm .0$ | $.08 \pm .03$ |
| BART | $.23 \pm .0$ | $.02 \pm .00$ | $.25 \pm .0$ | $.08 \pm .03$ |
| k-NN | $.23 \pm .0$ | $.02 \pm .01$ | $.26 \pm .0$ | $.13 \pm .05$ |
| RF | $.23 \pm .0$ | $.03 \pm .01$ | $.28 \pm .0$ | $.09 \pm .04$ |
| CF | $.19 \pm .0$ | $.03 \pm .01$ | $.20 \pm .0$ | $.07 \pm .03$ |
| BNN | $.20 \pm .0$ | $.04 \pm .01$ | $.24 \pm .0$ | $.09 \pm .04$ |
| TARNet | $.17 \pm .0$ | $.05 \pm .02$ | $.21 \pm .0$ | $.11 \pm .04$ |
| CFR-Wass | $.17 \pm .0$ | $.04 \pm .01$ | $.21 \pm .0$ | $.08 \pm .03$ |
| CEVAE | $.15 \pm .0$ | $.02 \pm .01$ | $.26 \pm .1$ | $.03 \pm .01$ |
| SITE | $.17 \pm .0$ | $.04 \pm .01$ | $.21 \pm .0$ | $.09 \pm .03$ |
| ABCEI | $.13 \pm .0$ | $.02 \pm .01$ | $.17 \pm .0$ | $.03 \pm .01$ |
| PSBT | $\mathbf{.10 \pm .0}$ | $.01 \pm .01$ | $\mathbf{.11 \pm .0}$ | $\mathbf{.02 \pm .01}$ |

Table 4: In-sample and out-of-sample results with mean and standard errors on the Twins dataset (AUC: higher = better, $\epsilon_{\text{ATE}}$: lower = better).

| Methods | In-sample | | Out-sample | |
|---|---|---|---|---|
| | AUC | $\epsilon_{\text{ATE}}$ | AUC | $\epsilon_{\text{ATE}}$ |
| OLS/$LR_1$ | $.660 \pm .005$ | $.004 \pm .003$ | $.500 \pm .028$ | $.007 \pm .006$ |
| OLS/$LR_2$ | $.660 \pm .004$ | $.004 \pm .003$ | $.500 \pm .016$ | $.007 \pm .006$ |
| S.Learner | $.680 \pm .009$ | $.111 \pm .013$ | $.520 \pm .033$ | $.131 \pm .015$ |
| T.Learner | $.695 \pm .008$ | $.091 \pm .008$ | $.580 \pm .024$ | $.105 \pm .009$ |
| BLR | $.611 \pm .009$ | $.006 \pm .004$ | $.510 \pm .018$ | $.033 \pm .009$ |
| BART | $.506 \pm .014$ | $.121 \pm .024$ | $.500 \pm .011$ | $.127 \pm .003$ |
| k-NN | $.609 \pm .010$ | $.003 \pm .002$ | $.492 \pm .012$ | $.005 \pm .004$ |
| BNN | $.690 \pm .008$ | $.006 \pm .003$ | $.676 \pm .008$ | $.020 \pm .007$ |
| TARNet | $.849 \pm .002$ | $.011 \pm .002$ | $.840 \pm .006$ | $.015 \pm .002$ |
| CFR-Wass | $.850 \pm .002$ | $.011 \pm .002$ | $.842 \pm .005$ | $.028 \pm .003$ |
| CEVAE | $.845 \pm .003$ | $.022 \pm .002$ | $.841 \pm .004$ | $.032 \pm .003$ |
| SITE | $.862 \pm .002$ | $.016 \pm .001$ | $.853 \pm .006$ | $.020 \pm .002$ |
| ABCEI | $.871 \pm .001$ | $.003 \pm .001$ | $.863 \pm .001$ | $.005 \pm .001$ |
| PSBT | $\mathbf{.885 \pm .001}$ | $\mathbf{.001 \pm .001}$ | $\mathbf{.876 \pm .001}$ | $\mathbf{.001 \pm .001}$ |

we know parts of the ground truth, and hence we can evaluate the precision of ATT estimation and policy risk estimation. Here:

$$R_{\text{pol}}(\pi) = 1 - \mathbb{E}\left(y_1 | \pi\left(x^u\right) = 1\right) \cdot P(\pi = 1) - \mathbb{E}\left(y_0 | \pi\left(x^u\right) = 0\right) \cdot P(\pi = 0). \qquad (1)$$

We consider $\pi(x^u) = 1$ when $f(x^u, 1) - f(x^u, 0) > 0$.

For the Twins dataset, because we only know the observed treatment outcome for each unit, we follow Louizos et al. (2017) in using the Area Under the ROC Curve (AUC) as the evaluation metric.

## 4.4 RESULTS

Tables 2-4 list experimental results on each of the four datasets. It would be inappropriate to aggregate the statistical test results reported across these tables. Due to the varying availability of ground truth, different evaluation metrics are used for each dataset, making it unsuitable to combine these metrics into a single statistical hypothesis test. However, PSBT demonstrates superior performance in 15 out of 16 cases. This is evident not only from having the best results in the columns but also from often exhibiting non-overlapping empirical confidence intervals compared to the closest competitor. This provides strong evidence that PSBT represents a significant improvement over the current state of the art.

The Jobs and IHDP datasets have the smallest numbers of observations, the smallest numbers of covariates, and a pronounced imbalance between control and treatment group sizes (cf. Table 1). Here, PSBT achieves competitive performance against baselines. On datasets with more observa-

Table 5: In-sample and out-of-sample results with mean and standard errors on the MIMIC-III benchmark (lower = better).

| Methods | In-sample | | Out-sample | |
|---|---|---|---|---|
| | $\sqrt{\epsilon_{PEHE}}$ | $\epsilon_{ATE}$ | $\sqrt{\epsilon_{PEHE}}$ | $\epsilon_{ATE}$ |
| OLS/$LR_1$ | 7.1 ± .2 | .92 ± .15 | 8.2 ± .2 | .97 ± .15 |
| OLS/$LR_2$ | 2.7 ± .1 | .24 ± .11 | 3.3 ± .2 | .29 ± .13 |
| S.Learner | 2.2 ± .2 | .36 ± .09 | 2.8 ± .3 | .39 ± .09 |
| T.Learner | 1.8 ± .1 | .31 ± .13 | 2.1 ± .1 | .33 ± .15 |
| BLR | 7.3 ± .1 | .90 ± .09 | 8.5 ± .3 | .97 ± .09 |
| BART | 2.4 ± .2 | .31 ± .09 | 3.1 ± .2 | .37 ± .12 |
| k-NN | 2.8 ± .1 | .32 ± .11 | 3.6 ± .1 | .36 ± .11 |
| RF | 4.6 ± .3 | .88 ± .10 | 5.3 ± .3 | .89 ± .11 |
| CF | 4.1 ± .1 | .22 ± .13 | 4.9 ± .1 | .24 ± .14 |
| BNN | 2.5 ± .1 | .45 ± .11 | 3.3 ± .1 | .49 ± .11 |
| TARNet | 1.91 ± .0 | .25 ± .16 | 2.11 ± .1 | .31 ± .16 |
| CFR-Wass | 1.06 ± .0 | .19 ± .14 | 1.09 ± .0 | .21 ± .14 |
| CEVAE | 2.71 ± .0 | .23 ± .11 | 2.72 ± .0 | .23 ± .12 |
| SITE | 1.29 ± .0 | .21 ± .14 | 1.35 ± .0 | .25 ± .14 |
| ABCEI | .85 ± .0 | .11 ± .12 | .89 ± .0 | .12 ± .14 |
| PSBT | **.55 ± .0** | **.07 ± .05** | **.59 ± .0** | **.07 ± .04** |

tions, more covariates, and greater balance between control and treatment groups, PSBT consistently performs better.

Regression-based methods struggle with high generalization error due to treatment selection bias. Nearest neighbor-based methods address selection bias by considering unit similarity but fail to achieve global balance. Recent advances in domain adaptation have improved causal effect estimation but suffer from the imbalance between treatment and control groups. PSBT makes use of propensity guidance to supervise the representation learning model to learn the causal knowledge, enabling PSBT to make counterfactual prediction with fine tuned causal models. This makes PSBT outperform baseline methods.

## 5 CONCLUSIONS

Properly answering questions like "*What would patient's outcome be had they taken medication A?*" is one of the central issues of the causal effect inference problems. Traditional methods focus on tackling the confounding bias problem by covariate balancing, learning a balanced representation for the treatment and control groups. We propose a new framework **PSBT**: a **P**ropensity **S**imilarity guided **B**idirectional **T**ransformer model for causal effect inference. PSBT makes use of the propensity knowledge to supervise the representation learning in order to teach the model to learn the differences between the propensities between the two units. A bidirectional transformer model is trained by two supervising tasks: the one is to learn to predict the covariate features that are randomly masked, the other is to learn to predict the propensity similarity. By learning the propensity similarity, the model learns to disentangle the confounding factors. After the pre-training stage, we apply a fine-tuning stage to fine-tune the pre-trained propensity model into the causal model. An additional neural network layer is employed to enable the causal prediction task. By conducting multiple experiments on several real-world datasets, we demonstrate that PSBT significantly outperforms traditional and state-of-the-art baseline methods.

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

## A  RELATED WORK

Research on causal effect inference provides insights into the underlying data-generating processes and enables us to answer counterfactual questions. Only one response can be observed at the same time. The fundamental challenge in causal effect inference lies in the identifiability problem given certain data and assumptions Tian & Pearl (2002). Properly designed causal models are used to guarantee the identifiable causal effect during the inference process Imbens & Rubin (2015). In order to satisfy the strong ignorability condition for unbiasedly estimate the causal effects, *Randomized Controlled Trials* (RCTs) are designed to create comparable groups for treatment effect estimation Rosenbaum (2002). In observational studies, these groups are achieved by matching units from different groups to meet the identifiability condition, which lead to the formulation of *Average Treatment effects on Treated* (ATT) Nikolaev et al. (2013). On the other hand, learning-based algorithms are developed to estimate the *Average Treatment Effects* (ATE) to achieve a comprehensive understanding about the causal effects on the population and individual level Chipman et al. (2008); Shalit et al. (2017).

Matching-based methods aim to create comparable units from treated and untreated groups, achieving locally balanced distributions. Techniques Wu et al. (2023) vary in their similarity measures. Propensity score matching Rosenbaum & Rubin (1983) is a notable example, using estimated propensity scores to assess similarity between units. Tree-based methods Wager & Athey (2018), which employ adaptive similarity measures, are also a part of this category, though they are often computationally intensive and challenging to apply in large-scale settings.

As opposed to matching-based methods, we employ the propensity model to estimate the distance between covariate pairs in the latent space, so that the knowledge of propensity could be learned by the representation learning model to ensure the identifiability condition. Propensity scores are often estimated using logistic regression models Chen et al. (2021); Dai et al. (2022); Lee et al. (2021), with techniques such as feature selection Wang et al. (2023a;b). A key example of an unbiased estimator is the inverse propensity score method Rosenbaum & Rubin (1983), which reweights each unit inversely to its estimated propensity score. However, this method can suffer from high variance in cases of low propensity and may introduce bias when propensity estimates are inaccurate Li et al. (2023a). To address these issues, doubly robust estimators and variance reduction techniques have

been developed Li et al. (2023b), although these methods are still limited by their dependence on propensity scores, impacting their practical effectiveness.

Learning-based methods attempt to map data into a feature space where distributional discrepancies are minimized. The primary challenge is accurately measuring these discrepancies. Initial studies employed metrics like maximum mean discrepancy and basic Wasserstein discrepancy Johansson et al. (2016); Shalit et al. (2017), while subsequent work introduced techniques such as local similarity preservation Yao et al. (2018; 2019), feature selection Cheng et al. (2022); Hassanpour & Greiner (2019), representation decomposition Wu et al. (2022), and adversarial training Yoon et al. (2018). Despite their effectiveness, these methods struggle in certain common scenarios, such as outlier presence Fatras et al. (2021) and unlabeled confounders, which can compromise the reliability of discrepancy measures.

One of the core issues in representation learning is to learn a meaningful information encoding for different modes of the input data. Contrastive information coding is one way to achieve this goal in unsupervised and semi-supervised learning scenarios. In general, contrastive learning aims to formulate a learning target by sampling similar pairs of data plus dissimilar ones. This allows for the representation to be formulated as clusters of latent classes Saunshi et al. (2019). In a causal inference scenario, these latent clusters can be related to the stratifications of covariate representations, which is indicated by the propensity models. We propose to achieve this goal by formulating sequential pairs and learn the distinguished representations in the transformer framework.

