# OpenReview forum: "Propensity Guided Transformer For Causal Effect Inference"
_ICLR.cc/2026/Conference — ICLR 2026 Conference Withdrawn Submission_

### Official Review · Reviewer_QhzP · 2025-10-22

**Soundness:** 3
**Presentation:** 1
**Contribution:** 3
**Rating:** 4
**Confidence:** 3

**Summary:**

This paper proposes a model for conditional average treatment effect prediction. It relies on learning representations that are predictive of propensity score similarities. It outperforms alternatives on experiments.

**Strengths:**

- Original approach.

- Experiments are thorough and demonstrate a real (statistically significant) edge of the method over baselines.

**Weaknesses:**

- The model is generally hard to follow, there are no equations or maths describing the model's components and the training algorithm.

- The paper looks hastily written: references don't have parentheses or brackets, some letter variables are written as plain-text variables and not letters.

- The questions given at the start of experiments do not seem to be exactly answered in the text commenting the results.

- The classification for the baselines is somewhat weird: all "matching" methods are also statistical methods, and among them, only k-NN is a matching method. Also, Crump et al (2009) is not the best reference for it...

**Questions:**

Can you address the above points?

In particular,

- What are the "learning target" and the "*metric for measuring the propensity similarity" in Section 3.1

- What does "$P(X_0) \circ P(X_1)$" mean in Figure 2?

- Can you clarify what are the tokens in your method when covariates are tabular (as it usually is in causal inference datasets)?

---

### Official Review · Reviewer_mjRE · 2025-10-24

**Soundness:** 2
**Presentation:** 1
**Contribution:** 2
**Rating:** 2
**Confidence:** 3

**Summary:**

The authors propose a new approach for CATE estimation. More specifically, they propose to pre-train a transformer in a BERT-style by predicting (a) masked features and (b) the similarity in propensity for two samples.
This pre-training strategy is, to my very best knowledge, novel.
On the IHDP, Jobs, Lalonde, MIMIC-III and Twins datasets, the authors provide very strong results in terms of ATE and CATE estimation for their method compared to various baselines.

**Strengths:**

I find the general idea to use BERT-style pre-training for the purpose of causal effect estimation very interesting; this does seem to be a very promising approach for estimating causal effects.

Furthermore, the core idea is clearly communicated to readers familiar with both BERT (style) pertaining as well as causal effect estimation.

Additionally, the results on the results on the IHDP, Jobs, Lalonde, MIMIC-III and Twins datasets look very strong.

**Weaknesses:**

The presentation of the paper is unfortunately not adequate for ICLR. One cannot help but feel like this is a very rushed submission.

This includes:

(a) the writing that is partially confused and unnecessarily convoluted in parts.
(b) Lack of formulas, especially regarding the loss for predicting the propensity similarity of two units. This crucial part of the paper remains very unclear.
(c) Basic formatting issues, including: The citations are never in parentheses, the figures are too small and their resolution is too low
(d) Furthermore, crucial details are missing, such as details on all baselines, details on training duration and setup, as well as details on the experiments.
(f) Additionally, the related literature does not cover actually related work on using tabular transformer networks for causal effect estimation, for instance Do-PFN (https://arxiv.org/abs/2506.06039) or Causal-PFN (https://arxiv.org/abs/2506.07918).

Additionally, the experiments are insufficient. This includes:
(a) Missing baselines, especially recent transformer-based methods, such as Do-PFN (https://arxiv.org/abs/2506.06039) and Causal-PFN (https://arxiv.org/abs/2506.07918)
(b) No tuning, i.e. hyper-parameter optimization for any baselines
(c) Reporting of training-compute and runtimes for all methods
(d) Ablations: For instance investigating the effectiveness of the two different pre-training objectives.

There is also no theoretical analysis of the paper included at the moment.

**Questions:**

How exactly does the propensity-similarity loss work?
Please also feel free to reply to any of the weaknesses mentioned above.

---

### Official Review · Reviewer_TKia · 2025-10-28

**Soundness:** 3
**Presentation:** 3
**Contribution:** 3
**Rating:** 6
**Confidence:** 2

**Summary:**

This paper proposes a novel framework called Propensity Similarity guided Bidirectional Transformer (PSBT) to estimate causal effects from observational data while addressing confounding bias. The PSBT adopts a pre-training and fine-tuning paradigm. The authors evaluate PSBT on four benchmark datasets using metrics like Precision in Estimation of Heterogeneous Effect, Average Treatment Effect, Average Treatment Effect on the Treated, and Area Under the ROC Curve. Experimental results show that PSBT outperforms traditional and state-of-the-art causal inference methods in 15 out of 16 evaluation scenarios.

**Strengths:**

- The pre-training-fine-tuning structure aligns with modern deep learning practices, making the model easy to implement and extend. The combination of self-supervised and weakly supervised tasks in pre-training is well-motivated, as it explicitly integrates causal prior knowledge into representation learning, rather than relying solely on covariate balancing.
- PSBT consistently outperforms 15 baseline methods on key metrics. The reported results are accompanied by standard errors, enhancing the credibility of the performance claims.

**Weaknesses:**

- Limited Discussion on Propensity Score Estimation Details: The paper mentions using a "multi-layer fully connected neural network with BatchNormalization and ReLU layers" to estimate propensity scores for pre-training, but provides no details on hyperparameters (e.g., number of layers, hidden unit size), training procedures (e.g., loss function, optimization algorithm), or how the propensity model’s accuracy impacts PSBT’s final performance. This lack of transparency makes it harder to replicate the results.
- Insufficient Analysis of Computational Complexity: As a Transformer-based model, PSBT likely has higher computational costs (e.g., training time, memory usage) compared to lightweight baselines (e.g., OLS, k-NN). The paper does not report runtime, GPU memory requirements, or scalability to larger datasets (beyond 25k observations), which is a critical consideration for real-world deployment.
- Superficial Exploration of Confounding Bias Robustness: While PSBT aims to address confounding bias, the paper does not test its performance under scenarios with unmeasured confounders or noisy proxy confounders—common challenges in observational studies. It remains unclear how PSBT would perform when the "strong ignorability" assumption (Assumption 1) is partially violated.

**Questions:**

- For the propensity score model used in pre-training: Could you provide specific hyperparameters (e.g., network depth, learning rate) and training details (e.g., batch size, number of epochs)? Additionally, how does the accuracy of the propensity model (e.g., AUC of propensity score prediction) correlate with PSBT’s final causal effect estimation performance?
- Can you supplement computational results, such as training time per epoch on each dataset, GPU memory usage, and scalability tests on larger datasets (e.g., 100k+ observations)? This would help readers assess PSBT’s practicality for large-scale observational studies.
- The paper assumes all confounders are measurable (Assumption 1). Have you conducted any ablation studies to test PSBT’s robustness when this assumption is relaxed—for example, by adding unmeasured confounders or noisy proxies to the datasets? If not, could you discuss potential modifications to PSBT to handle such scenarios?
- In the pre-training phase, why did you choose to mask 10% of covariate tokens (with 80% masked, 10% randomly replaced, 10% unchanged)? Was this ratio selected through hyperparameter tuning, and how would changing the masking ratio (e.g., 5% or 20%) affect PSBT’s performance?

---

### Official Review · Reviewer_dzyi · 2025-11-01

**Soundness:** 1
**Presentation:** 1
**Contribution:** 1
**Rating:** 0
**Confidence:** 4

**Summary:**

The authors propose a bidirectional transformer to represent covariate features in a manner that accounts for propensity model so representations can be subsequently used for causal estimation without the need for learning balanced representations.

**Strengths:**

None.

**Weaknesses:**

This is a very poorly written paper that proposes learning a representation model for causal inference by making it of propensity scores learned separately. The main three issues with the presentation are that i) there are not enough details in Section 3, Figure 1 and Figure 2 to describe how the model and optimization objectives are actually defined, ii) there is not (at least an intuitive) justification for the model to address and mitigate observed confounding, and iii) although the results indicate that the proposed approach is competitive with existing approaches (after accounting for variation), without implementation details of which there are none, it is extremely difficult to take them at face value.

**Questions:**

It is the opinion of the reviewer that the paper will have to be almost completely rewritten to make it suitable for publication at ICLR or any other top machine learning venue. Consequently, it is advisable that the authors make an effort to better describe and justify their approach and present enough implementation details to make the results easier to understand and assess relative to other methods in the state-of-the-art.

---

### Note · Authors · 2025-11-19

I have read and agree with the venue's withdrawal policy on behalf of myself and my co-authors.